**DOI: 10.1038/ncomms14876**　　**OPEN**

# A redox-mediated Kemp eliminase

Aitao Li[1,2,*], Binju Wang[3,*], Adriana Ilie[1,2], Kshatresh D. Dubey[3], Gert Bange[4], Ivan V. Korendovych[5], Sason Shaik[3] & Manfred T. Reetz[1,2]

The acid/base-catalysed Kemp elimination of 5-nitro-benzisoxazole forming 2-cyano-4-nitrophenol has long served as a design platform of enzymes with non-natural reactions, providing new mechanistic insights in protein science. Here we describe an alternative concept based on redox catalysis by P450-BM3, leading to the same Kemp product via a fundamentally different mechanism. QM/MM computations show that it involves coordination of the substrate's N-atom to haem-Fe(II) with electron transfer and concomitant N–O heterolysis liberating an intermediate having a nitrogen radical moiety Fe(III)–N• and a phenoxyl anion. Product formation occurs by bond rotation and H-transfer. Two rationally chosen point mutations cause a notable increase in activity. The results shed light on the prevailing mechanistic uncertainties in human P450-catalysed metabolism of the immuno-modulatory drug leflunomide, which likewise undergoes redox-mediated Kemp elimination by P450-BM3. Other isoxazole-based pharmaceuticals are probably also metabolized by a redox mechanism. Our work provides a basis for designing future artificial enzymes.

[1] Department of Biocatalysis, Max-Planck-Institut für Kohlenforschung, Kaiser-Wilhelm-Platz 1, Mülheim an der Ruhr 45470, Germany. [2] Department of Chemistry, Philipps-Universität Marburg, Marburg 35032, Germany. [3] Institute of Chemistry and the Lise Meitner-Minerva Center for Computational Quantum Chemistry, The Hebrew University of Jerusalem, Jerusalem 9190401, Israel. [4] LOEWE Center for Synthetic Microbiology (SYNMIKRO) and Department of Chemistry, Philipps-Universität Marburg, Marburg 35032, Germany. [5] Department of Chemistry, Syracuse University, 111 College Place, Syracuse, New York 13244, USA. * These authors contributed equally to this work. Correspondence and requests for materials should be addressed to S.S. (email: sason.shaik@gmail.com) or to M.T.R. (email: reetz@mpi-muelheim.mpg.de).

The Kemp elimination[1] of 5-nitrobenzisoxazole has become a *de facto* experimental springboard for designing new protein catalysts to uncover the principles that govern enzymatic catalysis. For more than two decades[2], Kemp eliminases based on various protein scaffolds have provided valuable insights into understanding and mimicking enzymes[2–15] (Fig. 1a). Combining computational design methodology with 17 rounds of state-of-the-art directed evolution Hilvert, Mayo and coworkers reported the artificial enzyme HG3.17, which employs a Brønsted acid/base mechanism to catalyse Kemp elimination with an unprecedented catalytic efficiency that approaches the activities of natural enzyme ($k_{cat} = 700\,s^{-1}$ and $k_{cat}/K_m = 230{,}000\,M^{-1}\,s^{-1}$)[16]. While these numbers are truly impressive, analysis of the possible efficiency limits for base-catalysed Kemp elimination shows that additional improvement is still possible in principle[17]. Along a different line, it has been postulated that an oxidoreductase breaks down 5-nitrobenzisoxazole possibly by an oxidative process, although no mechanistic evidence was reported[18].

In our work, we wanted to identify a catalyst that employs a mechanism beyond acid–base mechanism, specifically a redox process. From a fundamental point, finding a protein scaffold that catalyses the Kemp elimination by a fundamentally different mechanism would not only further our understanding of the function, genesis and evolution of enzymes, but also provide new opportunities for creating catalysts for novel chemical transformations.

Since enzymes that facilitate redox reactions are quite abundant, we speculated that a redox-based mechanism should be possible in Kemp elimination. We focused on cytochrome P450 monooxygenases (CYPs) for several reasons: Human cytochrome P450 (CYP) enzymes play a crucial role in the metabolism of therapeutic drugs[19–21], including the degradation of a number of prominent isoxazole-based pharmaceuticals[22–24]. A prime example is the metabolism of leflunomide, an anti-inflammatory agent used in the treatment of rheumatoid arthritis[23]. The main metabolic outcome is the physiologically active teriflunomide (2-cyano-3-oxo-N-[(4-trifluoromethyl)phenyl]butyramide, also called A771726), a formal Kemp elimination product. Two pathways have been postulated, an acid/base and an undefined redox-mediated process[22,23], but mechanistic studies to distinguish between the two possibilities have not been reported to date. The term Kemp elimination was not referred to in this connection. CYPs are remarkably promiscuous, lending themselves to non-natural reactions as shown, for example, by the seminal report of Dawson, Breslow

and coworkers describing CYP as a catalyst in inter- and intramolecular nitrene CH insertion[25] and by the recent discovery of related reactions that can be catalysed by this class of enzymes[26]. Moreover, rat liver microsomes that contain CYPs were reported to catalyse the dehydration of aldoximes, either Lewis acid/Brønsted base catalysis or some kind of a redox process being postulated[27]. Clearly, in these biological transformations mechanistic ambiguities persist to this day.

Here we show on the basis of experimental and computational data that the biocatalytic Kemp elimination of 5-nitro-benzisoxazole with formation of 2-cyano-4-nitrophenol need not proceed by the traditional acid/base mechanism. Cytochrome P450 monooxygenase from *Bacillus megaterium* (P450-BM3) and rationally designed mutants constitute active Kemp eliminases that indeed follow a redox-mediated mechanism. This finding has ramifications regarding the human metabolism of the immuno-modulatory therapeutic drug leflunomide and other isoxazole-based pharmaceuticals.

## Results

**Kemp elimination activity test with wild-type P450-BM3.** We speculated that the ferrous haem cofactor in the self-sufficient P450-BM3 (refs 21,28) could coordinate to the most basic position of substrate **1** at the isoxazole nitrogen. An internal redox process was then expected in which Fe(II) is oxidized to Fe(III) by electron-flow to the electron-deficient 5-nitrobenzisoxazole nucleus with concurrent rupture of the weakest bond (O–N), followed by proton transfer and formation of the formal Kemp elimination product **2** (Fig. 1b).

Wildtype (WT) P450-BM3 rapidly converts substrate **1** into **2** in a cell free extract (CFE) in the presence of NADPH for keeping the haem cofactor reduced. To exclude the possibility that an unknown protein in the CFE may catalyse this Kemp elimination, the enzyme was purified and tested, and the product identity was confirmed by gas chromatography–mass spectrometry (GC–MS) analysis (Supplementary Figs 1 and 2). Kinetic experiments using purified enzyme showed that activity of WT P450-BM3 in the Kemp elimination is remarkable. The observed turnover number ($k_{cat}/K_m = 240 \pm 60\,s^{-1}\,M^{-1}$; $k_{cat} > 1.5\,s^{-1}$; $K_m > 6\,mM$, exact $k_{cat}$ and $K_m$ values are only estimated due to low substrate solubility) is higher than those of any of the previously designed catalysts (including catalytic antibodies) before the application of directed evolution, as well as serum albumins[4].

**Figure 1 | Mechanisms of two different types of Kemp eliminases. (a)** Base-mediated exothermic E2 elimination. **(b)** A possible redox-mediated process.

**Experimental evidence for the redox mechanism.** A number of control experiments were designed to support the mechanistic hypothesis of a redox process. To begin with, experiments done with the purified enzyme showed that in the absence of NADPH no reaction occurs, while the addition of carbon monoxide (CO) to the reaction mixture resulted in a 10-fold loss of activity (Supplementary Table 1). Moreover, conducting the reaction under anaerobic conditions led to an almost twofold activity improvement (Supplementary Table 2). These findings are consistent with competition of CO or oxygen with substrate **1** in coordinating to ferrous haem. Thus, they are not compatible with a classical acid/base mechanism. On this basis we concluded that haem-Fe(II) is essential for the P450-BM3 catalysed Kemp elimination, and that a redox mechanism is likely. Further strong evidence for a redox mechanism was gained by substituting haem-Fe(II) in WT P450-BM3 with haem-Zn(II) according to a well-tested and reliable procedure[29] (Supplementary Figs 3 and 4). P450-BM3 containing Zn(II) failed to catalyse the Kemp elimination of substrate **1**, while the activity was fully recovered by the reconstituted protein with Fe-porphyrin IX (hemin) (Supplementary Table 3). Here again, the results are incompatible with an acid/base mechanism.

Evidence for a redox mechanism was also gained by a mutational experiment. It is well known that a cysteine-to-serine mutation[30] in P450-BM3 at the axial Fe-binding position C400S eliminates monooxygenation activity but enhances the efficiency of $Fe^{III}$-to-$Fe^{II}$ reduction by using NADPH as a reductant[26]. Therefore, we tested P450-BM3 variant C400S in the Kemp elimination of **1**. As anticipated on the basis of a redox mechanism, it showed an almost sixfold improvement in the catalytic efficiency ($k_{cat}/K_m$) relative to WT P450-BM3 ($k_{cat}/K_m = 1,400 \pm 150\,s^{-1}\,M^{-1}$; $k_{cat} = 3.8 \pm 0.4\,s^{-1}$; $K_m = 2.7 \pm 0.4\,mM$; $k_{cat}/k_{uncat} = 3.3 \times 10^6$).

**Mechanistic evidence by MD/(QM/MM) computations.** To elucidate the mechanistic details of the redox-mediated Kemp elimination in WT P450-BM3, we performed molecular dynamic (MD) simulations and quantum mechanics/molecular mechanics (QM/MM) calculations. These computations revealed no evidence for a pose of substrate **1** in which a classical acid/base mechanism is possible (Fig. 2). In addition, a π–π interaction between the phenyl group of phenylalanine (F87) and the substrate was identified, which sterically prevents tight binding of **1** with haem-Fe(II) necessary for optimal redox reaction (Fig. 3a). MD simulations for the oxidized haem state showed that the substrate never approach the ferric haem moiety (Supplementary Fig. 5). As such, the key reactive species is the ferrous and not the ferric haem species.

Based on a well-tested and reliable procedure for metalloenzymes[31–33], QM/MM calculations were performed by using the equilibrated snapshots from the MD simulations of the Fe(II) haem state (see Supplementary Methods for details). Figure 3b presents the QM/MM calculated energy profile for the Kemp elimination of **1** in WT P450-BM3. As the MD simulation reveals (Fig. 3a), the substrate has an upright orientation, in which its N-atom is directed towards the Fe(II) center, while the H-atom is pointing down toward the haem. Moreover, no base residue is identified in the vicinity of this H to accept the substrate proton. Starting from the initial reactant complex of **1**, the attack of the substrate's N-atom onto Fe(II) is coupled with electron transfer from the latter, thus leading to a heterolytic cleavage of N–O bond. No computational evidence for a short-lived Lewis acid/Brønsted base complex was evident (Supplementary Fig. 7), which is partially because the haem is reduced (negatively charged), and there is significant electrostatic repulsion between

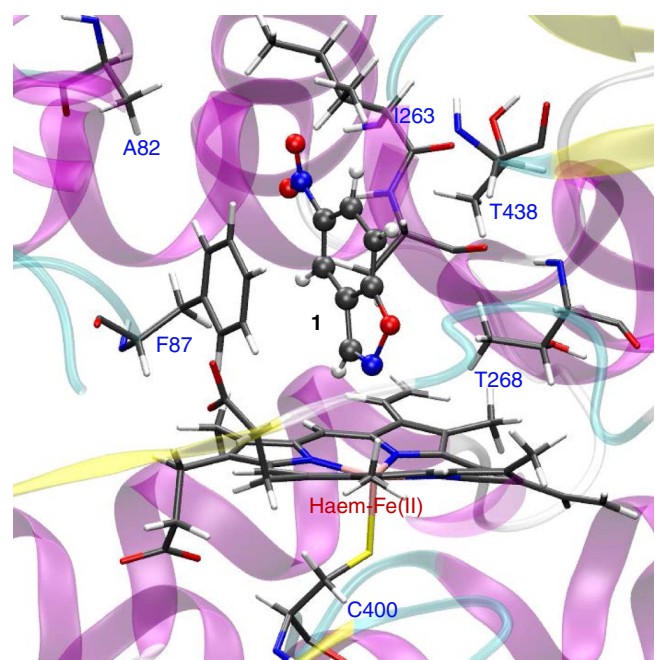

**Figure 2 | Overview of substrate 1 accommodated in the active site of WT P450-BM3.** The representative snapshot was taken from molecular dynamic (MD) simulation, revealing substrate **1** in the binding pocket of P450-BM3 and the surrounding residues. As can be seen, there are no residues which could act as in an acid/base mechanism.

the substrate's N and the haem moiety. According to the calculations, the electron transfer from Fe(II) to the substrate generates intermediate **IN1** (Fig. 3b), wherein the N–O bond is broken, and the haem iron is bonded to the nitrogen to form the nitrogen radical moiety Fe(III)–N• and a phenoxyl anion. The spin density at N is −0.62, indicating notable spin localization at this atom. Subsequent bond rotation places H close to O via **TS2** with formation of intermediate **IN2** (Fig. 3b). Mulliken charge analysis reveals the transfer of a significant charge (−0.74e) from Fe(II) to the substrate in **TS2**, and as such we are dealing with a redox-mediated mechanism. The final proton transfer to $O^-$ via **TS3** leads to the final product and regeneration of Fe(II) (Fig. 3b). The reactions involve two-state reactivity[34,35]. The reaction starts with the quintet ground state of Fe(II), but crosses to the triplet state already at the first step of the reaction. The calculated overall barrier (via $^3$**TS2**) with zero-point energy correction is $21.1\,kcal\,mol^{-1}$.

For comparison, we also investigated the reactivity of Zn(II)-haem with QM/MM calculations. Our QM/MM calculations show the substrate activation via Zn(II)-haem experiences a huge barrier (over $50\,kcal\,mol^{-1}$ in Supplementary Fig. 8). Moreover, no stable intermediate like **IN1** could be located. This result is consistent with the experimental findings and redox-mediated process in Fe(II)-haem, since Zn(II) is usually not a redox center. Finally, we compared the positive change on Fe(II) in Fe(II)-haem and Zn(II) in Zn(II)-haem with QM/MM calculations, and found the positive charge on Zn(II) (+0.74e) is even bigger than that in Fe(II) (+0.66e), this excludes the possibility that Fe(II) plays a purely electrostatic role in the concerted elimination reaction and further supports the redox-mediated mechanism.

**Engineering P450-BM3 for activity improvement.** Led by these computational predictions, we turned to mutational studies. Since optimal binding of substrate **1** to ferrous haem appeared to be

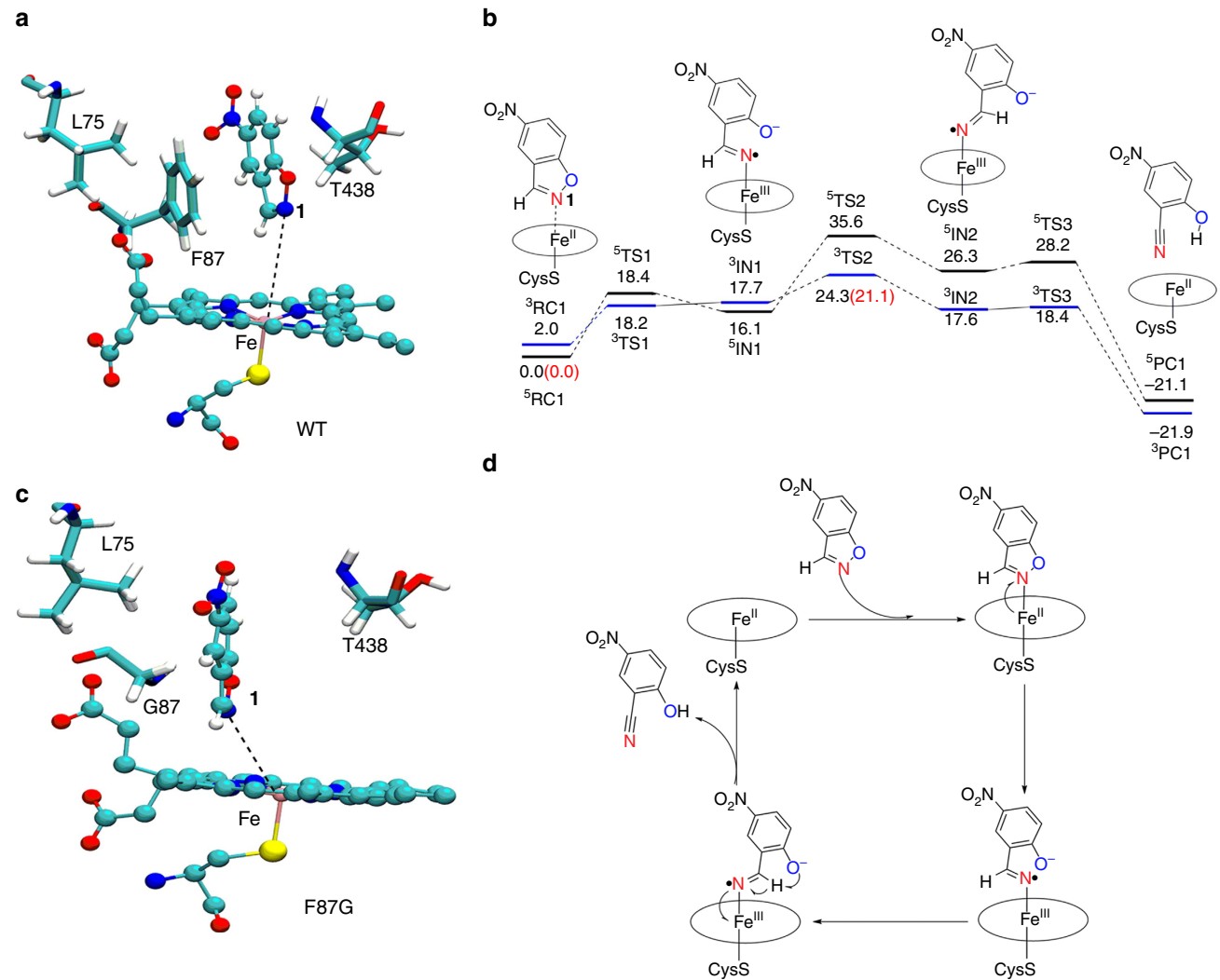

**Figure 3 | Redox-mediated Kemp elimination of substrate 1.** (**a**) A representative snapshot in the equilibrium molecular dynamic (MD) trajectory showing the active site structure of WT P450-BM3; note that the nitrogen atom of substrate **1** is directed toward Fe(II), with an average Fe—N1 distance of 4.44 Å. (**b**) Quantum mechanics/molecular mechanics (QM/MM) (UB3LYP/B2) relative energies (kcal mol$^{-1}$) for the redox-mediated Kemp elimination of **1**. All values are dispersion-corrected. The values in parentheses also include zero-point energy (ZPE) corrections. For clarity, the reactivity energy profile in the singlet state is provided in Supplementary Fig. 6. Cartesian coordinates of QM region for all species from QM/MM calculations are available as Supplementary Data 1. (**c**) A representative snapshot in the equilibrium MD trajectory showing the active site structure of variant F87G, with an average Fe—N1 distance of 4.11 Å. (**d**) Catalytic cycle for redox-mediated Kemp elimination.

**Table 1 | Summary of kinetic parameters for P450-BM3 and variants catalysing the Kemp elimination of 1.**

| Catalyst* | $K_m$ (mM) | $k_{cat}$ (s$^{-1}$) | $k_{cat}/K_m$ (s$^{-1}$ M$^{-1}$) | $k_{cat}/K_{uncat}$[†] |
|---|---|---|---|---|
| WT P450-BM3 | >6 | >1.5 | 240 ± 60 | 1.3 × 10$^6$ |
| C400S | 2.7 ± 0.4 | 3.8 ± 0.4 | 1,400 ± 150 | 3.3 × 10$^6$ |
| F87G | 2.1 ± 0.4 | 2.7 ± 0.3 | 1,300 ± 140 | 2.3 × 10$^6$ |
| F87A | 2.0 ± 0.3 | 1.2 ± 0.1 | 600 ± 50 | 1.0 × 10$^6$ |
| F87V | 0.50 ± 0.04 | 1.1 ± 0.1 | 2,200 ± 200 | 0.9 × 10$^6$ |
| F87I | >4 | >1.5 | 320 ± 40 | 1.3 × 10$^6$ |
| A82F | 0.27 ± 0.03 | 8.4 ± 0.4 | 31,000 ± 1,500 | 7.0 × 10$^6$ |
| F87G/A82F | 1.3 ± 0.2 | 11.5 ± 0.7 | 8,800 ± 700 | 10.0 × 10$^6$ |
| A82F/C400S | 0.61 ± 0.13 | 2.9 ± 0.4 | 4,800 ± 700 | 2.5 × 10$^6$ |

*Assay conditions: 25 °C, 50 mM sodium phosphate buffer pH 8.0, 100 mM NaCl, 0.25 mM NADPH, 100 nM or 300 nM purified enzyme, 5% acetonitrile. Errors correspond to ± s.d. determined from at least three independent measurements.
†Rate accelerations for the Kemp eliminases were calculated based on the rate constant for the uncatalyzed reaction determined in ref. 10 ($k_{uncat} = 1.16 \times 10^{-6}$ s$^{-1}$).

prevented sterically by phenylalanine at residue F87, we introduced smaller amino acids at this position. As listed in Table 1, mutant F87I shows comparable activity as WT P450-BM3 due to

similar size of Ile and Phe. The $K_m$ values for mutants F87V, F87A and F87G are significantly lowered, and the highest activity was observed for F87G. The catalytic efficiency ($k_{cat}/K_m$) of this

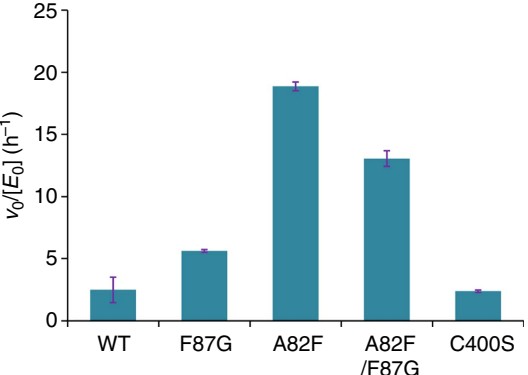

**Figure 4 | Metabolism of leflnomide with human P450.** Human P450 catalyses the isoxazole ring scission of leflunomide, an immunomodulatory therapeutic drug[22–24], leading to teriflunomide (A771726).

variant is more than fivefold higher relative to WT P450-BM3 due to the lower Michaelis constant $K_m$ and improved activity $k_{cat}$ (Table 1). The subsequent MD simulations (Fig. 3c) revealed that the Fe(II)–N$_{benzisoxazole}$ distance in variant F87G is indeed shorter than in the case of the WT enzyme (see Supplementary Fig. 9 for details).

Next, we attempted to reshape the large substrate binding pocket of P450-BM3 to better match the small size of the substrate. The A82F mutation in P450-BM3 was previously shown to greatly enhance the enzyme's binding affinity for small molecules, leading to significant improvements in catalytic hydroxylation efficiency[36]. Therefore, we introduced the A82F mutation into both WT P450-BM3 and F87G P450-BM3. Kinetic characterization of the two resulting variants showed 129- and 36-fold improvements in catalytic efficiencies ($k_{cat}/K_m$) in the A82F and A82F/F87G mutants, respectively, due to both improved turnover number and the Michaelis constant (Table 1). These results underscore the predictive power of the combination of MD and QM/MM with empirical observations that allowed the identification of Kemp eliminase activity based on a novel redox-mediated mechanism (Fig. 3d). The turnover number observed for the rationally designed A82F mutant obtained after screening only three additional variants is higher than in any of the previously designed Kemp eliminases with the exception of HG3.17 (ref. 16).

**Metabolism of leflunomide by P450-BM3.** Following the work with the model compound **1**, we turned to leflunomide as substrate, knowing that this compound is metabolized by human P450 (refs 22–24) (Fig. 4). As anticipated, WT P450-BM3 as well as selected variants A82F, F87G, C400S and A82F/F87G proved to be active with formation of the formal Kemp elimination product teriflunomide (A771226) (Fig. 5, Supplementary Figs 10 and 11). In full agreement with the results obtained for **1**, the A82F mutant is ~8-fold more active in metabolizing leflunomide as compared to WT P450-BM3. With the exception of variant C400S, they all led to a small amount of side-product, likely to be the hydroxylation product as reported earlier in human P450-catalysed metabolism of leflunomide[37]. Based on our experimental and computational results using P450-BM3 as the catalyst in the reaction of the standard substrate **1**, the same redox mechanism may be postulated in human metabolism of leflunomide, but these mechanistic details still need to be explored. Relevant is also the reported human metabolism of a different isoxazole-based therapeutic drug, zonisamide[23], active in the treatment of epilepsy, Parkinson's disease, dardive dyskinesia, migraine and obesity. The metabolite is a ring-opened compound, namely 2-(sulfamoylacetyl)phenol (Supplementary Fig. 12), which cannot be formed by a classical acid/base-mediated Kemp elimination since it lacks a hydrogen at the 3-position. In this case a redox mechanism could be involved, but this needs to be checked in future research.

**Figure 5 | Activity test of P450-BM3 variants in metabolizing leflunomide.** Reaction conditions: 500 μl reaction consisting of 1 μM enzyme, 500 μM substrate and 1 mM NADPH in phosphate buffer (50 mM, pH 8.0, 100 mM NaCl), 25 °C, 1,000 r.p.m. for 1 h. All reactions were performed in triplicate, and error bars show ± s.d.

## Discussion

Various protein scaffolds have been reported as Kemp eliminases, all operating by the traditional acid/base mechanism[2–17]. In contrast, we have identified a Kemp eliminase that functions by a fundamentally different mechanism based on a redox process. WT P450-BM3 and several rationally designed mutants are capable of efficiently catalysing ring-opening of the standard model compound 5-nitro-benzisoxazole **1** with formation of the formal Kemp product 2-cyano-4-nitrophenol **2**. Extensive MD and QM/MM calculations coupled with mechanistic and mutational studies clearly point to a redox-mediated bond cleavage mechanism as opposed to the conventional acid/base process. The experimental evidence is summarized by the following points: in the absence of NADPH no reaction occurs; addition of CO impedes activity; reaction under anaerobic conditions results in twofold activity improvement; especially mutants A82F, F87G, C400S, A82F/F87G and A82F/C400S lead to improvements in catalytic efficiency; the haem-Zn(II) analogue of WT P450-BM3 shows no activity in the standard reaction of substrate **1**.

WT P450-BM3 catalyses the Kemp elimination via the redox path with an efficiency ($k_{cat}/K_m = 240 \pm 60 \, \text{s}^{-1} \, \text{M}^{-1}$) that is higher than those of any previously designed Kemp eliminases in the absence of mutations introduced by directed evolution, as well as serum albumins[4]. Several rationally designed mutants resulted in pronounced rate enhancement, variant A82F showing the best performance ($k_{cat}/K_m = 31,000 \pm 1,500 \, \text{s}^{-1} \, \text{M}^{-1}$), which corresponds to a $k_{cat}/k_{uncat}$ value of $7.0 \times 10^6$. Further improvements can be expected by applying directed evolution, or possibly by modifying the metal centre[29].

Importantly, we have also shown that WT P450-BM3 and mutants thereof catalyse the transformation of the therapeutic drug leflunomide with formation of the formal Kemp product teriflunomide (A771226) (Figs 4 and 5). This transformation was known to be involved in the metabolism of leflunomide in humans, but its mechanism has remained elusive[22]. We believe that the redox mechanism shown in Fig. 3b will inspire respective research of human P450-mediated metabolism of leflunomide and of other isoxazole-based pharmaceuticals such as zonisamide[23], hopefully ending any remaining mechanistic ambiguities. Finally, a variety of redox-catalysed promiscuous reactions can be envisioned for practical applications, extending the list of P450-mediated non-natural reactions[26].

## Methods

**PCR based method for variants creation.** Variants were created using the QuikChange protocol with Hot Start DNA polymerase from *Thermococcus koda-karaensis*. An aliquot of 50 μl reaction mixtures typically contained 30 μl water, 5 μl DNA polymerase buffer ( × 10), 3 μl 25 mM MgSO₄, 5 μl 2 mM dNTPs, 2.5 μl DMSO, 0.5 μl (50–100 ng) template DNA, 0.5 μl for each primer (100 μM) and 1 μl KOD hot start polymerase. The PCR conditions are as follows: 95 °C 3 min, (95 °C 30 s, 60 °C 30 s, 68 °C 5 min 30 s) × 24 cycles, 68 °C 10 min, 16 °C 30 min. The PCR products were analysed on agarose gel by electrophoresis and purified using a Qiagen PCR gel extraction kit. A total of 2 μl NEB CutSmart Buffer and 2 μl *Dpn* I were added in 50 μl PCR reaction mixture and the digestion was carried out at 37 °C for 6 h. After *Dpn* I digestion, the PCR products (1 μl) were directly transformed into electro-competent *Escherichia coli* BL21(DE3) to create the variants for screening. All primers used are listed in Supplementary Table 4.

**Protein production and purification.** The variants selected were produced and purified for biochemical characterization. *E. coli* BL21(DE3) cells were transformed with the plasmids prsfDuet-1 (Novagen)[38] containing the gene of interest. To ensure monoclonality, single-colony streakouts were first prepared and inoculated into 4 ml LB medium containing 50 μg ml⁻¹ kanamycin and cultured overnight at 37 °C, 220 r.p.m. The overnight culture (4 ml) was transferred into 200 ml TB with 50 μg ml⁻¹ kanamycin in 500 ml shaking flasks. The cultivation continued at 37 °C, 220 r.p.m. for 2–3 h until the OD₆₀₀ reached 0.6–0.8, then IPTG was added to a final concentration of 0.2 mM and the temperature was reduced to 25 °C. After 20 h of expression, the cells were collected by centrifugation at 4,000 r.p.m., 4 °C for 15 min. The cell pellets were stored at −80 °C until further processing. The cell pellets were disrupted by sonication and the tube was kept in an ice bath during sonication. The collected lysate was centrifuged for 45 min at 11,000 r.p.m. at 4 °C and the obtained brownish-red supernatant was filtered to sterility with a 0.45 μm filter. The lysate was loaded onto a nickel affinity column (GE Healthcare) and washed with 10–250 mM imidazole solution containing 800 mM NaCl and 50 mM potassium phosphate buffer (pH 8.0). Proteins from the flow through were pooled and concentrated, and then desalted using Hitrap desalting column equilibrated with 100 mM potassium phosphate buffer (pH 8.0). A flow rate of 5 ml min⁻¹ was used and all fractions showing adsorption at 417 nm were collected and concentrated to a final volume of 1 ml with Amicon Ultra centrifugal filters (cutoff 50 kDa). The protein was shock frozen with liquid nitrogen and stored at −80 °C until further usage (see Supplementary Fig. 13 for protein SDS–PAGE analysis).

**Determination of enzyme concentration.** Total enzyme concentration including the active P450 and inactive forms P420 (no monooxygenation activity) was determined by CO difference spectrum analysis[39] for both purified enzyme and CFE. The extinction coefficients of $\varepsilon_{450-490} = 91,000\,M^{-1}\,cm^{-1}$ and $\varepsilon_{420-490} = 110,000\,M^{-1}\,cm^{-1}$ were used for the P450 and cytochrome P420, respectively. For the serine ligated mutants P411, an extinction coefficient of $\varepsilon_{411-490} = 103,000\,M^{-1}\,cm^{-1}$ was employed[40]. Kemp elimination activity was measured based on the total enzyme concentration since the enzyme without monooxygenation activity is still active towards the Kemp substrate.

**Substrate solubility.** Solubility of 5-nitrobenzisoxazole (substrate **1**) under the assay conditions (50 mM sodium phosphate buffer with 100 mM NaCl, pH 8.0, 5% acetonitrile (ACN), 25 °C) was quantified according to the previously reported method[16]. The solubility limit of 5-nitrobenzisoxazole under these conditions was determined to be 3.2 mM.

**Kinetic assay.** The assays were conducted on a JASCO V-650 spectrophotometer with quartz cuvettes monitoring absorbance at 380 nm at 25 °C using at least three independent measurements. In a typical experiment the purified enzyme (with final concentration ranging from 0.1 to 0.3 μM) was added to freshly prepared 5-nitrobenzisoxazole substrate (50 μM–2 mM final concentration) in 50 mM sodium phosphate buffer (pH 8.0) containing 100 mM NaCl and 5% ACN. The reactions were initiated by adding NADPH with a final concentration of 0.25 mM. The slope before addition of NADPH was subtracted as background. Product's extinction coefficient (15,800 cm⁻¹ M⁻¹) was used[8]. Initial rates divided by catalyst concentration were plotted against substrate concentration, and $k_{cat}$ and $K_m$ values were determined by fitting the data to the Michaelis–Menten equation $v_0/[E]_0 = k_{cat}[S]/(K_m + [S])$. In the case of WT P450-BM3 and variant-F87I-catalysed cleavage of 5-nitrobenzisoxazole, the individual of $k_{cat}$ and $K_m$ values could not be determined due to limited substrate solubility, only the enzymatic efficiency ($k_{cat}/K_m$) could be measured by fitting the linear portion of the Michaelis–Menten graph to the following equation: $v_0/[E]_0 = (k_{cat}/K_m)[S]$, the $K_m$ and $k_{cat}$ values were estimated based on the linearity of the graph. The kinetic data is presented in Table 1 and Supplementary Fig. 14.

**Activity assay under anaerobic conditions.** Stock solutions of substrate in acetonitrile, NADPH and P450-BM3 in phosphate buffer (50 mM, pH 8.0, 100 mM NaCl) were degassed separately by removing the air and flushing argon into

solution. In a typical experiment, into a 1 ml spectrophotometer quartz cuvette with a sealing lid (that was previously prepared for anaerobic conditions by removing the air and adding argon using vacuum line), was sequentially added purified protein (945 μl with concentration of WT 1 μM), substrate (50 μl in ACN, final concentration 0.5 mM) and then NADPH (5 μl, final concentration 0.25 mM). All three components were carefully added using Hamilton syringes in the cuvette that was continuously flushed with argon via a needle. The cuvette was sealed and immediately subjected to activity assay. To investigate the effect of CO on activity, the enzyme solution saturated with CO was employed. All assays were performed at least in three independent measurements.

**Preparation of reconstituted P450-BM3 protein.** The directed expression, Ni-NTA purification, and metalation of apo-P450-BM3 protein to generate the reconstituted P450-BM3 proteins containing haem-Fe (hemin) or haem-Zn(II) were conducted by strictly following the well-tested and reliable procedure[29]. The protein concentration was measured using the Bradford assay. Ultraviolet–vis spectra of reconstituted proteins were tested using a JASCO V-650 spectrophotometer.

**MD simulations.** The initial structures of P450-BM3 were taken from Protein Data Bank with PDB code of 1JPZ (ref. 28). The substrate 5-nitrobenzisoxazole was docked into the active site of P450-BM3 using AutoDock Vina tool[41] in Chimera[42]. The force field for the haem moiety in the resting state (Fe(III)) was taken from the literature[43], while the force field for the one-electron reduced state (Fe(II)) was parameterized using 'MCPB.py' model[44]. The general AMBER force field[45] was used for the substrate 5-nitrobenzisoxazole, while the partial atomic charges and missing parameters for the substrate were obtained from the RESP model[46], using HF/6–31G* level of theory. After proper minimizations and equilibrations, a productive MD run of 100 ns was performed for each system. The Amber ff14SB force field[47] was employed for the protein in all of the MD simulations. All MD simulations were performed with GPU version of Amber 14 package[48] (see Supplementary Methods for computational system preparation and setup; as well as other computational details for MD simulations).

**QM/MM calculations.** Equilibrated snapshots from the MD simulations were taken for the subsequent QM/MM calculations. All QM/MM calculations were performed using ChemShell[49,50], combining Turbomole[51] for the QM part and DL_POLY[52] for the MM part. The MM region was treated by CHARMM27 force field[53], while QM region with 58 atoms was treated by the hybrid UB3LYP functional[54] with two basis sets. For geometry optimization and frequency calculations the all electron basis set of def2-SVP[55], referred to as B1, was used. The energies are further corrected with the large all-electron basis-set Def2-TZVP (ref. 55), labelled as B2. The empirical dispersion energy correction was calculated for all species by using the DFT-D3 programme[56] (see Supplementary Methods for more computational details of QM/MM calculations). For comparison, a bigger QM region with 73 atoms was also tested for the first reaction step of Kemp elimination of **1**, and the result is not much changed (Supplementary Fig. 15).

**Metabolism of leflunomide with WT P450-BM3 and variants.** Reaction mixture of 500 μl typically contained 1 μM enzyme, 500 μM leflunomide, 2% (v/v) methanol as co-solvent and 1 mM NADPH in phosphate buffer (50 mM, pH 8.0, 100 mM NaCl). The reaction was incubated at 25 °C, 1,000 r.p.m. for 1 h. After reaction, ACN 500 μl was added to quench the reaction. The supernatant was obtained by centrifugation at 12,000 r.p.m. for 5 min, filtered and then subjected to HPLC analysis.

**HPLC analysis.** The quantitative assessments of lefunomide and its Kemp product teriflunomide (A771726) were analysed on a Shimadzu LC-2010CHT system with a column of Zorbax Eclipse 5 XDB-C18 250 × 4.6 mm (Agilent Technologies, Palo Alto, CA) at 40 °C and 254 nm. A binary gradient consisting of a mixture of 10 mM ammonium formate, 0.1% formic acid (solvent A) and ACN (solvent B) at a flow rate of 1 ml min⁻¹ was employed. The LC gradient was programmed as follows: solvent A to solvent B ratio was held at 100:0 (v/v) for 3 min and then adjusted from 100:0 (v/v) to 10:90 (v/v) for 20 min and from 10:90 (v/v) to 100:0 (v/v) from 20 to 25 min. Leflunomide and teriflunomide were separated with a retention time of 15.3 and 20.4 min, respectively.

**Chemistry.** The details can be found in Supplementary Information (see Supplementary Figs 16 and 17; Supplementary Methods for chemicals and GC–MS analysis).

**Data availability.** The crystal structure used to perform the MD simulation studies is archived from RCSB Protein Data Bank with PDB code of 1JPZ. All other data supporting the findings of this study are available within the paper and its Supplementary Information Files.

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

## Acknowledgements

M.T.R. thanks the Max-Planck-Society (reference number 106705) and the LOEWE Research cluster SynChemBio (reference number 56150029) for generous support.

S.S. acknowledges support from the Israel Science Foundation (ISF grant 1183/13), and B.W. and K.D.D. are supported in part by a PBC fellowship. G.B. thanks the LOEWE initiative of the state of Hessen for support, and I.V.K. acknowledges support from the Alexander von Humboldt Foundation (reference number 61508772) and the NIH (grant GM 119634).

## Author contributions

A.L. and A.I. performed the experiments; B.W. and K.D.D. performed the MD simulations and QM/MM study; the project is supervised by M.T.R. and S.S.; A.L., B.W., S.S., G.B., I.V.K. and M.T.R. wrote the manuscript; all authors checked the manuscript.

## Additional information

**Competing interests:** The authors declare no competing financial interests.

