## [Peer Review File · Nature Communications]

Reviewer #1 (Remarks to the Author):

This paper reports that P450-BM3 catalyzes the elimination reaction of 5-nitro-benzisoxazole to form 2-cyano-4-nitrophenol, the effect of several site-directed mutations on the kinetic parameters for the protein-catalyzed reaction, and the results of QM/MM calculations to model one proposed reaction mechanism. The author's claim that their results provide strong support for a novel redox-mediated Kemp elimination, rather than the well documented concerted base-catalyzed elimination reaction mechanism

This work is not ready for publication for several reasons.

(1) Many proteins - most notably bovine serum albumin - catalyze the Kemp elimination reaction, because effective catalysis is achieved simply by holding the substrate close to an active site base at a nonpolar protein binding site. All of these conditions might hold for P450-BM3, and the authors need to consider the possibility that coordination of the substrate nitrogen to the Fe(II) site activates the substrate for base catalyzed elimination through electrostatic stabilization of transition state negative charge, particularly at the oxygen leaving group. This mechanism differs from that in Figure 1b, because it requires a base at the enzyme active site to accept the substrate proton.

The authors write on the bottom of page 5 "concurrent rupture of the weakest bond (O-N), followed by proton transfer and formation of the formal Kemp elimination product 2 (Fig. 1b)". I strongly recommend that they add a formal discussion of the proton acceptor at the enzyme active site. If this were an active site base for a concerted Kemp elimination reaction, then the mechanism would look similar to that for Figure 1a. The major point of contention then reduces to whether Fe(II) plays a purely electrostatic role in stabilization of the transition state for the concerted elimination reaction, or if instead there is some (small?) amount of formal electron transfer from iron to nitrogen at the transition state. The experiments with the Zn(II) heme are useful in this regard. The authors should comment upon the relative nitrogen affinities of Fe(II) and Zn(II) hemes.

(2) The experimental results reported in this paper demonstrate an essential role for the heme cofactor, but otherwise provide little insight into reaction mechanism. The authors should therefore emphasize that they have no experimental evidence for the formation of a nitrogen radical intermediate, in which there is formal electron transfer from Fe(II) to nitrogen.

(3) The authors write "QM/MM computations show that it involves coordination of the substrate's N atom to heme-Fe(II) with electron transfer and concomitant N-O heterolysis liberating an intermediate having a nitrogen radical moiety Fe(III)-N• and a phenoxyl anion." The quality of these these QM/MM calculations at an enzyme active site is simply not sufficient to justify this sweeping conclusion. Similar calculations by a properly motivated chemist, with a similar belief in his or her mechanistic intuition, might also support the concerted elimination reaction described above.

Reviewer #2 (Remarks to the Author):

The authors present an interesting case of reaction promiscuity in human P450: catalysis of Kemp elimination of isoxazoles by a redox-based mechanism. Overall, the work is solid and convincing, and my comments and suggestions are listed below.

1) The idea of Kemp elimination catalysis by a redox mechanism is not absolutely novel. During a screen of ASKA library (a collection of over expressed E Coli genes), putative oxidoreductase was previously found to catalyze Kemp elimination of 5-nitrobenzisoxazole (Khersonsky et al,

Biochemistry 2011). In this work, however, the characterization of redox reaction is much more thorough.

2) Mutational analysis:

a) The authors make F87G mutation that reduces steric hindrance and indeed improves the activity. Have other identities been tried? Phe>Gly is a very drastic change and perhaps intermediate hydrophobic residues would give a more fine-tuned effect.

b) Are there any ideas about the non-additivity of mutations A82F and F87G?

c) Why C400S mutation was not recombined with other mutations that improve the activity?

3) The comparison of P450 proficiency with that of de novo designed Kemp eliminases is not very relevant, since the authors describe a natural enzyme with a promiscuous Kemp eliminase activity, improved by rational design. It would be better to compare the activity of P450 with that of BSA, which also exhibits Kemp eliminate activity (Hollfelder et al, Nature 1996).

4) Since serum albumin, which is extremely abundant in human serum, can also perform Kemp Elimination, the authors should check whether it can metabolize leflunomide before claiming that leflunomide metabolism occurs by redox mechanism (lines 218-219).

Reviewer #3 (Remarks to the Author):

General:

The authors take a well-known model enzymatic reaction, the usually base-catalyzed Kemp elimination on 5-nitro-benzisoxazole to 3-cyanonitrophen-4-ol, and demonstrate that the reagent can be reacted to the same end product via redox catalysis with cytochrome P 450 BM3, a well-known monooxygenation catalyst. The authors also demonstrate redox catalysis on another isoxazole, leflunomide, to teriflunomide, the ring-opened beta-cyano-methyl ketone product. The work is important because it demonstrates the existence of alternative pathways to get from substrate to product. The work has been conducted carefully and the experiments have been backed up by QM/MM calculations. After minor corrections spelled out below, the work should be published.

Detail:

1. Zonisamide, lines 221 ff.: While the rest of the manuscript deals with fact-based items, very good reasoning, and robust reliable experimental results, the section on zonisamide seems speculative and thus out of place. Owing to methylenesulfonamide substitution in 5-position, zonisamide does not necessarily mechanistically resemble substrate 1 or Leflunomide. This section should be struck.
2. Figure 5: the x-axis labelling should read: 'Cyt P450-BM3 variants'. The term 'conversion' in the caption is wrong, the y-axis shows an enzymatic rate constant. However, the maximum conversion should be included in Figure 5 or elsewhere in the text.

Clerical:

- i) Figure 2: leflunomide is misspelled in the caption; instead of 'A771726', teriflunomide should be used.
- ii) Table 1: last column should read 'kcat/kuncat'. All entries in that column should have the same exponent, not two with 6 and two with 7.

Reviewer #4 (Remarks to the Author):

The manuscript "A redox-mediated Kemp eliminase" by Aitao Li et al. identifies a redox-based mechanism for the Kemp elimination of 5-nitrobenzisoxazole by the enzyme P450-BM3. The proposed mechanism involves coordination of the substrate nitrogen to the ferrous heme cofactor followed by electron transfer from the heme iron to nitrogen with N-O bond rupture and formation of Fe(III) and an N-radical intermediate, rotation around the adjacent C-C bond, intra-substrate proton transfer, and formation of the formal Kemp elimination product. This mechanism is supported by computational studies (MD and QM/MM) and by a range of experiments of the WT enzyme and mutational experiments. Based on previous knowledge and insights from the simulations, 4 mutants were expressed with point mutations in 3 different sites. The authors demonstrate that with this rational design approach, the activity of the enzyme is increased by 6 to 139-fold. The manuscript also demonstrates that WT P450-BM3 and the mutants catalyze Kemp elimination of the drug molecule leflunomide. This is important since it suggests that other drugs are metabolized following the same mechanism, for instance zonisamide, which has no hydrogen in 3-position for a base-catalyzed mechanism. These findings are novel and should be of interest to others in the community.

The manuscript is well written and most relevant experimental and computational details are given in the Supporting Information. I am not able to comment on the quality and appropriateness of the experimental protocols, however, the work seems to be carried out with care and is well documented. The computational models seem appropriate, although a more thorough discussion of the selection of the QM region for the QM/MM calculations should be given. The results of QM/MM calculations are sensitive to selection of residues that are included in the QM region. I am confident that the authors have made a sensible choice and the major conclusions would likely not change by increasing the QM region size, but I would urge the authors to discuss and justify the selection in the Methods Section. This information will be useful for others in the field, in particular since none of the amino acid residues in vicinity of the heme group (apart from the coordinating cysteine) were included.

The SI needs to be checked. I visualized structure 5RC1, and it does not look like the coordinating RC1 structure, it looks like a snapshot from the MD in which the substrate is not coordinating. Structure 5TS1 looks like it could be the first transition state, so it may be just wrong coordinates for 5RC1.

The citations in the Methods section for MD simulations and QM/MM calculations are very selective and incomplete. E.g. why are ChemShell, Turbomole and DL_POLY cited but not Autodock and AMBER? Why are the heme FF parameters and MCBP.py cited but not the ff14SB and CHARMM27 force fields? I suggest to add citations for AutoDock, the AMBER MD code the authors used (GPU code as I gather from SI), ff14SB and CHARMM force fields, B3LYP functional, and Karlsruhe def2 basis sets.

Which force field was used for the substrate 5-nitrobenzisoxazole? I assume it is GAFF, but the information is missing. Please add and cite corresponding reference.

I wonder why in Table 1 no value is given for the rate acceleration of the WT enzyme. If $k_{cat} > 1.5$, it should be $> 1.3 \times 10^6$.

Reviewer #1 (Remarks to the Author):

This paper reports that P450-BM3 catalyzes the elimination reaction of 5-nitro-benzisoxazole to form 2-cyano-4-nitrophenol, the effect of several site-directed mutations on the kinetic parameters for the protein-catalyzed reaction, and the results of QM/MM calculations to model one proposed reaction mechanism. The author's claim that their results provide strong support for a novel redox-mediated Kemp elimination, rather than the well documented concerted base-catalyzed elimination reaction mechanism.

This work is not ready for publication for several reasons.

(1) Many proteins - most notably bovine serum albumin - catalyze the Kemp elimination reaction, because effective catalysis is achieved simply by holding the substrate close to an active site base at a nonpolar protein binding site. All of these conditions might hold for P450-BM3, and the authors need to consider the possibility that coordination of the substrate nitrogen to the Fe(II) site activates the substrate for base catalyzed elimination through electrostatic stabilization of transition state negative charge, particularly at the oxygen leaving group. This mechanism differs from that in Figure 1b, because it requires a base at the enzyme active site to accept the substrate proton.

We have considered the possibility of the mechanism suggested by the reviewer in the very beginning. However, there is no available base in the active site to accept the substrate proton. As clearly shown in the equilibrated snapshot from long-time scale MD simulations (Page 7, Figure 3), the substrate proton is pointing down towards the heme plan, and no base residue is identified in the vicinity. Moreover, we cannot locate a species in which the substrate's N is coordinated to Fe(II) without N-O bond cleavage (see Supplementary Figure 8), this is partially because the heme is reduced (negatively charged), and there is significant electrostatic repulsion between substrate N and heme. Up to now, the only available activation pathway is redox-mediated mechanism, in which an electron shifted to the substrate, coupled with N-O cleavage. The redox mechanism is also supported by combined experimental and computational study on Zn-heme (see further reply to reviewer 1 below).

The authors write on the bottom of page 5 "concurrent rupture of the weakest bond (O-N), followed by proton transfer and formation of the formal Kemp elimination product 2 (Fig. 1b)". I strongly recommend that they add a formal discussion of the proton acceptor at the enzyme active site. If this were an active site base for a concerted Kemp elimination reaction, then the mechanism would look similar to that for Figure 1a. The major point of contention then reduces to whether Fe(II) plays a purely electrostatic role in stabilization of the transition state for the concerted elimination reaction, or if instead there is some (small?) amount of formal electron transfer from iron to nitrogen at the transition state. The experiments with the Zn(II) heme are

useful in this regard. The authors should comment upon the relative nitrogen affinities of Fe(II) and Zn(II) hemes.

In addition to our reply to the first question, we want to stress that our calculations do not support the notion that Fe(II) plays a purely electrostatic role, as the substrate's N does not coordinate to Fe(II) without the N-O cleavage. Instead, our QM/MM calculations demonstrate a clear redox mechanism as in Figure 4. In the rate-determining TS2, there is significant charge transfer from Fe(II) to the substrate (-0.74e from Fe(II) to the substrate). For comparison, we also investigated the reactivity of Zn(II) heme with QM/MM calculations. Our QM/MM calculations show the substrate's activation via Zn(II)-heme experiences a huge barrier (>50 kcal/mol in Figure S9). Moreover, no stable intermediate like IN1 could be located. This new result is consistent with the experimental findings. More importantly, it indirectly supports the redox mechanism since Zn(II) is not a redox center in our common sense. Finally, we compared the positive change on Fe(II) in Fe(II)-heme and Zn(II) in Zn(II)-heme with QM/MM calculations, and found the positive charge on Zn(II) (+0.74e) is even higher than that in Fe(II) (+0.66e), this further excludes the purely electrostatic role of Fe(II). (See more discussion on page 9 in main text and Supplementary Figure 9 as new computational data)

(2) The experimental results reported in this paper demonstrate an essential role for the heme cofactor, but otherwise provide little insight into reaction mechanism. The authors should therefore emphasize that they have no experimental evidence for the formation of a nitrogen radical intermediate, in which there is formal electron transfer from Fe(II) to nitrogen.

It is true that we do not have direct experimental evidence for the nitrogen radical intermediate. However, we should recognize such nitrogen radical intermediate as IN1 (In Figure 4) is just an unstable intermediate involved in the reaction. In such a case, any techniques such as Electron spin resonance (ESR) would probably fail. In this scenario, the state-of-the-art QM/MM calculations are highly complementary to experimental work and are valuable tools for revealing the enzymatic mechanism and activity at the atomic level. As discussed above, our QM/MM calculations demonstrate a clear and novel redox-mediated mechanism for Kemp elimination. That is why we took great efforts in performing the above laboratory experiments in addition to carrying out very extensive state-of-the-art QM calculations.

(3) The authors write "QM/MM computations show that it involves coordination of the substrate's N atom to heme-Fe(II) with electron transfer and concomitant N-O heterolysis liberating an intermediate having a nitrogen radical moiety Fe(III)-N• and a phenoxyl anion." The quality of these these QM/MM calculations at an enzyme active site is simply not sufficient to justify this sweeping conclusion. Similar calculations by a properly motivated chemist, with a similar belief in his or her mechanistic intuition, might also support the concerted elimination reaction described above.

We disagree with this conjecture. The QM/MM approach we used is well tested and has proven in hundreds of papers and reviews to be reliable for metalloenzymes, especially for P450 systems. The mechanism demonstrated in the paper is revealing as discussed above.

Moreover, we have performed additional QM computations as summarized in Supplementary Figures 6, 8 and 9, showing the results in the early phase of the reaction with a larger QM region, and in the triplet and singlet states, respectively, for comparison (see also response to reviewer 4 below).

Reviewer #2 (Remarks to the Author):

The authors present an interesting case of reaction promiscuity in human P450: catalysis of Kemp elimination of isoxazoles by a redox-based mechanism. Overall, the work is solid and convincing, and my comments and suggestions are listed below.

1) The idea of Kemp elimination catalysis by a redox mechanism is not absolutely novel. During a screen of ASKA library (a collection of over expressed E Coli genes), putative oxidoreductase was previously found to catalyze Kemp elimination of 5-nitrobenzisoxazole (Khersonsky et al, Biochemistry 2011). In this work, however, the characterization of redox reaction is much more thorough.

In the above reference, an oxidoreductase was mentioned as breaking down 5-nitrobenzisoxazole, but mechanistic evidence was not presented. We have now cited this paper accordingly. (page 3, lines 13-15, reference 18)

2) Mutational analysis:

a) The authors make F87G mutation that reduces steric hindrance and indeed improves the activity. Have other identities been tried? Phe>Gly is a very drastic change and perhaps intermediate hydrophobic residues would give a more fine-tuned effect.

We have now created more mutants F87I, F87V and F87A. The corresponding kinetic data is shown in Table 1 in the revised manuscript. Mutant F87I shows the similar results with the WT, since Ile has a similar size as Phe, both mutants showed relative large K_m . Two other new mutants F87V and F87A were also generated, showing much lower K_m values due to decreasing steric hindrance (the substrate N atom is getting closer to the heme-Fe(II)), however, the activity (k_{cat}) ($1.2 \pm 0.1 \text{ s}^{-1}$ for F87 A and $1.1 \pm 0.1 \text{ s}^{-1}$ for F87V) is much lower compared to mutant F87G ($2.7 \pm 0.3 \text{ s}^{-1}$). These new data support our original mechanistic conclusion, and therefore the discussion has been extended in the revised manuscript (See page 10, lines 1-9).

b) Are there any ideas about the non-additivity of mutations A82F and F87G?

In terms of activity, it is approximately additive for the mutations A82F and F87G, the k_{cat} values of the three mutants F87G, A82F and F87G/A82F are 2.7 S^{-1} , 8.4 S^{-1} and 11.5 S^{-1} , respectively. $2.7 \text{ S}^{-1} + 8.4 \text{ S}^{-1} = 11.1 \text{ S}^{-1}$ which is quite close to the value 11.5 S^{-1} . However, non-additivity was observed in terms of catalytic efficiency (k_{cat}/K_m), since K_m value of mutant A82F is very small (0.27 mM), when the mutation F87G was further introduced based on A82F, the volume of binding pocket was broadened and K_m value became above 2 times bigger than the single mutant A82F (meaning much more substrate needed to reach saturated state) which caused the even lower k_{cat}/K_m of mutant F87G/A82F compared with A82F.

However, we don't think that this is relevant to our study.

c) Why C400S mutation was not recombined with other mutations that improve the activity?

We have now combined A82F and C400S with formation of the double mutant A82F/C400S. Activity is not increased as shown in the new Table 1.

3) The comparison of P450 proficiency with that of de novo designed Kemp eliminases is not very relevant, since the authors describe a natural enzyme with a promiscuous Kemp eliminase activity, improved by rational design. It would be better to compare the activity of P450 with that of BSA, which also exhibits Kemp eliminate activity (Hollfelder et al, Nature 1996).

The sentence reflecting the comparison of P450 with BSA for the Kemp elimination has been added (Page 5, line 15).

4) Since serum albumin, which is extremely abundant in human serum, can also perform Kemp Elimination, the authors should check whether it can metabolize leflunomide before claiming that leflunomide metabolism occurs by redox mechanism (lines 218-219).

We have tested the activity with human serum albumin for the substrate leflunomide and found that it is also active, as already reported (*Drug Metab. Dispos.* **31**, 1240–1250 (2003), our reference 22). It most likely operates by a classical acid/base mechanism. However, this is not relevant to our work which focuses on P450 enzymes. As we state in our original paper and in the revision, metabolism of leflunomide with P450 is well documented (*Drug Metab. Dispos.* **31**, 1240–1250 (2003), but the mechanism had not been elucidated. Our present work points the way.

Reviewer #3 (Remarks to the Author):

General:

The authors take a well-known model enzymatic reaction, the usually base-catalyzed Kemp elimination on 5-nitro-benzisoxazole to 3-cyanonitrophen-4-ol, and demonstrate that the reagent can be reacted to the same end product via redox catalysis with cytochrome P 450 BM3, a well-known monooxygenation catalyst. The authors also demonstrate redox catalysis on another isoxazole, leflunomide, to teriflunomide, the ring-opened beta-cyano-methyl ketone product. The work is important because it demonstrates the existence of alternative pathways to get from substrate to product. The work has been conducted carefully and the experiments have been backed up by QM/MM calculations. After minor corrections spelled out below, the work should be published.

Detail:

1. Zonisamide, lines 221 ff.: While the rest of the manuscript deals with fact-based items, very good reasoning, and robust reliable experimental results, the section on zonisamide seems speculative and thus out of place. Owing to methylenesulfonamide substitution in 5-position, zonisamide does not necessarily mechanistically resemble substrate 1 or Leflunomide. This section should be struck.

We have rewritten this section, pointing out that a redox mechanism may be involved but that this needs further mechanistic work (See page 11, lines 23-25).

2. Figure 5: the x-axis labelling should read: 'Cyt P450-BM3 variants'. The term 'conversion' in the caption is wrong, the y-axis shows an enzymatic rate constant. However, the maximum conversion should be included in Figure 5 or elsewhere in the text.

We have modified the figure 5 accordingly; the maximum conversion is also given in the Supplementary Information (Supplementary Figure 12).

Clerical:

i) Figure 2: leflunomide is misspelled in the caption; instead of 'A771726', teriflunomide should be used.

We have modified the figure 2 accordingly.

ii) Table 1: last column should read 'k_{cat}/k_{uncat}'. All entries in that column should have the same exponent, not two with 6 and two with 7.

We have corrected them accordingly in the Table 1 in the revised manuscript.

Reviewer #4 (Remarks to the Author):

The manuscript "A redox-mediated Kemp eliminase" by Aitao Li et al. identifies a redox-based mechanism for the Kemp elimination of 5-nitrobenzoxazole by the enzyme P450-BM3. The proposed mechanism involves coordination of the substrate nitrogen to the ferrous heme cofactor followed by electron transfer from the heme iron to nitrogen with N-O bond rupture and formation of Fe(III) and an N-radical intermediate, rotation around the adjacent C-C bond, intra-substrate proton transfer, and formation of the formal Kemp elimination product. This mechanism is supported by computational studies (MD and QM/MM) and by a range of experiments of the WT enzyme and mutational experiments. Based on previous knowledge and insights from the simulations, 4 mutants were expressed with point mutations in 3 different sites. The authors demonstrate that with this rational design approach, the activity of the enzyme is increased by 6 to 139-fold. The manuscript also demonstrates that WT P450-BM3 and the mutants catalyze Kemp elimination of the drug molecule leflunomide. This is important since it suggests that other drugs are metabolized following the same mechanism, for instance zonisamide, which has no hydrogen in 3-position for a base-catalyzed mechanism. These findings are novel and should be of interest to others in the community.

The manuscript is well written and most relevant experimental and computational details are given in the Supporting Information. I am not able to comment on the quality and appropriateness of the experimental protocols, however, the work seems to be carried out with care and is well documented. The computational models seem appropriate, although a more thorough discussion of the selection of the QM region for the QM/MM calculations should be given. The results of QM/MM calculations are sensitive to selection of residues that are included in the QM region. I am confident that the authors have made a sensible choice and the major conclusions would likely not change by increasing the QM region size, but I would urge the authors to discuss and justify the selection in the Methods Section. This information will be useful for others in the field, in particular since none of the amino acid residues in vicinity of the heme group (apart from the coordinating cysteine) were included.

According to the suggestion of the referee, the choice of the QM region was added in the Methods section in Supplementary Information (See SI, Page 8). The QM region in our QM/MM calculations contained 58 atoms, including the heme without side chains, the coordinating cysteine and the substrate. For the surrounding standard amino acid residues, they are lying away from the substrate and do not form strong electronic interactions (such as salt bridge, strong H-bonding interactions or involving electron transfer) with the selected QM region (heme moiety and the substrate), so that the standard MM charge and force field are sufficient for our

QM/MM study. For comparison, we also tested the first reaction step on a larger QM region with 73 atoms, in which the two closest residues were included (the phe87 and a water as shown in Supplementary Figure 6, see SI, Page 17). The QM/MM results in Supplementary Figure 6 indicate that the overall barrier is not much affected by the QM region size (18.2 kcal/mol in Figure 4 vs 16.6 kcal/mol in Supplementary Figure 6)

The SI needs to be checked. I visualized structure 5RC1, and it does not look like the coordinating RC1 structure, it looks like a snapshot from the MD in which the substrate is not coordinating. Structure 5TS1 looks like it could be the first transition state, so it may be just wrong coordinates for 5RC1.

We have checked and it is the correct initial reactant complex. A stable reactant complex in which the substrate is coordinated to Fe(II) is not located in our QM/MM calculations (Supplementary Figure 8, see SI, Page 18), this is partially because the heme is reduced (negatively charged), and there is significant electrostatic repulsion between substrate N and heme. As shown in our QM/MM calculations (Figure 4), the substrate activation is a concerted process, that is, the attack of substrate N onto the Fe(II) is coupled with N-O bond cleavage and electron transfer from Fe(II) to substrate. Nevertheless, a reactant complex with substrate N coordinating to Fe(II) without N-O bond cleavage cannot be located. In summary, the new QM computations reflected in Supplementary Figure 8 fully support our original conclusions .

The citations in the Methods section for MD simulations and QM/MM calculations are very selective and incomplete. E.g. why are ChemShell, Turbomole and DL_POLY cited but not Autodock and AMBER? Why are the heme FF parameters and MCBP.py cited but not the ff14SB and CHARMM27 force fields? I suggest to add citations for AutoDock, the AMBER MD code the authors used (GPU code as I gather from SI), ff14SB and CHARMM force fields, B3LYP functional, and Karlsruhe def2 basis sets.

According to the suggestions of referee, the relevant and missing references are cited in both main text and Supplementary Information (see references 38, 39, 42, 44, 45, 50-52 in the main text, references 8-10, 13, 16, 17, 27, 29, 31 in the Supplementary Information).

Which force field was used for the substrate 5-nitrobenzoxazole? I assume it is GAFF, but the information is missing. Please add and cite corresponding reference.

The referee is correct; the parameter of substrate is taken from GAFF.

I wonder why in Table 1 no value is given for the rate acceleration of the WT enzyme. If $k_{cat} > 1.5$, it should be $> 1.3 \times 10^6$.

We have added this value in the Table 1 in the revised manuscript.

Sincerely,

Andreas Goetz

Reviewer #1 (Remarks to the Author):

I continue to view these results as dubious, but difficult to challenge. Perhaps the base that abstracts the proton in a classical elimination reaction is a metal stabilized hydroxide anion. I don't know, and cannot make a compelling case against publication of this work. It has the "tone" of a Nature publication.

Reviewer #2 (Remarks to the Author):

The authors of the manuscript describing catalysis of Kemp elimination by cytochrome P450 via redox mechanism have answered all the points of my review. I therefore recommend to publish the revised manuscript.

Reviewer #4 (Remarks to the Author):

The authors have addressed all points that were raised by the reviewers. In my opinion the revised manuscript can be published.